# Monitoring Green Sea Turtles in the San Gabriel River of Southern California

**DOI:** 10.3390/ani13030434

**Published:** 2023-01-27

**Authors:** Lynn M. Massey, Shannon Penna, Eric Zahn, Dan Lawson, Cassandra M. Davis

**Affiliations:** 1West Coast Region, National Marine Fisheries Service, NOAA, Long Beach, CA 90802, USA; 2Tidal Influence, Long Beach, CA 90802, USA; 3Aquarium of the Pacific, Long Beach, CA 90802, USA

**Keywords:** green sea turtle, *Chelonia mydas*, citizen science, conservation, management, habitat, wetland

## Abstract

**Simple Summary:**

The East Pacific population of green sea turtles (*Chelonia mydas*) has undergone substantial growth in recent years, and as such, green sea turtle sightings are becoming more common along the U.S. West Coast. The northernmost resident population of green sea turtles in the eastern Pacific Ocean lives near the mouth of the San Gabriel River in Long Beach, California, USA. Utilizing nine years (2013–2021) of citizen science data from the Aquarium of the Pacific’s Southern California Sea Turtle Monitoring Project, we established a year-round presence of this population and determined that the areas along a 2.4-km (1.5 mile) stretch of the lower San Gabriel River with the most green sea turtle activity are near the Los Cerritos Wetlands and a power plant warm water effluent area, which are located approximately 1.3 and 2.9 km (0.8 and 1.8 miles), respectively, upriver from the mouth and entrance to Alamitos Bay. We hypothesize that turtles are attracted to these areas of the river for forage opportunity and thermal refuge. As green sea turtle presence in Southern California continues to increase, we recommend expanded monitoring programs to help understand essential habitat needs for this threatened population.

**Abstract:**

Effective conservation of endangered species relies on the characterization of habitat use and tracking of long-term population trends, which can be especially challenging for marine species that migrate long distances and utilize a diversity of habitats throughout their lives. Since 2012, citizen science volunteers at the Aquarium of the Pacific in Long Beach, California, have been monitoring an urban population of East Pacific green sea turtles (*Chelonia mydas*) that resides near the mouth of the San Gabriel River (SGR) in Southern California, USA, in order to gain insights about how the population uses this area. Here, we collate and analyze nine years of citizen science data, including observed sightings collected across 10 observation stations. Our results confirm that green sea turtles are frequently present around warm water effluent from power plants, similar to research results reported for other locations in the eastern Pacific Ocean. Importantly, observational data also show notable green sea turtle activity around the outfalls for a small wetland habitat bordering the SGR, highlighting the importance of wetland ecosystems as a key habitat and foraging area for this threatened population. Finally, our results showcase the benefits of using citizen science to monitor sea turtle populations in easily accessible nearshore habitats.

## 1. Introduction

Green sea turtles (*Chelonia mydas*) have been a species of conservation concern at a global level for decades [1]. Although green sea turtle populations throughout the world are rebounding [2,3,4,5], they have historically experienced significant population decline due to direct hunting [6], overharvesting of eggs [7], incidental fisheries bycatch [8,9,10,11], habitat loss, and climate change [12,13]. Efforts to restore populations are enhanced when greater information is available about their habitat usage and population status throughout their range. 

The East Pacific green sea turtle (hereafter referred to as “green turtle”) constitutes its own distinct population segment that is listed as threatened under the U.S. Endangered Species Act [14]. Green turtles can be found in coastal foraging areas from the U.S. West Coast to Chile, with nesting sites most densely concentrated in central Mexico, the Galapagos Islands, and Costa Rica [15]. Green turtles transit from these nesting sites to coastal and offshore areas throughout the eastern Pacific Ocean, including California, USA [16], where they forage on seagrass, macroalgae, and small invertebrates [17,18]. Juvenile green turtles from this population have been known to demonstrate high site fidelity, often remaining in the same foraging area for a decade or more while growing to maturity [17,19]. Over the last 25 years, this population has undergone a remarkable rebound, owed largely to successful protection of their primary nesting beaches and foraging areas in Mexico [15]. As a result, green turtle occurrence at the northern end of their range in Southern California has increased in recent years [20].

### 1.1. Natural History of Green Turtles in the San Gabriel River

In 2010, the presence of green turtles near the mouth of the San Gabriel River (SGR), located in Long Beach, California, was established through documentation of observations made by professional biologists and local community members, indicating that green turtles occur year-round in areas farther north than previously recorded on the U.S. West Coast [21,22]. Prior to this finding, a well-studied population of green turtles in San Diego Bay, located approximately 193 km (km, 120 miles (mi)) south of Long Beach, was thought to be the northernmost resident population in the eastern Pacific Ocean [23,24]. As ectothermic reptiles, green turtles typically occupy warmer waters in the tropics and subtropics to regulate their body temperatures [25,26]. Since green turtles prefer warmer waters, local biologists hypothesized that the turtles were taking thermal refuge in the SGR, as they were consistently sighted in and near warm water effluent from two electricity-generating plants [27]. Both of these power plants use a process called once-through-cooling (OTC) whereby ambient cold seawater is pumped into the plants to cool their steam generators and discharged to the SGR as warm water effluent [28,29]. This hypothesis was supported by Crear et al. [30] in a study that tracked the year-round movement of these green turtles via acoustic telemetry, which recorded a denser turtle presence around the power plants in the winter months when ambient river temperatures were colder. 

### 1.2. Citizen Science as a Tool for Sea Turtle Research

The term citizen science (CS), often referred to as community science [31], is most broadly defined as the involvement of the public in scientific research [32,33]. The CS framework often involves training volunteer non-scientists to collect data for scientific projects [34], which can then be compiled and analyzed for informing management decisions [35]. Such projects have provided vital data for monitoring and conserving wildlife populations [36,37,38,39]. 

CS can be particularly useful for research projects that track ecological patterns (e.g., population trends and habitat shifts) of organisms that are difficult to observe [40]. As with most marine megafauna, sea turtles are difficult to observe because they migrate long distances and spend large amounts of time submerged [41]. Sea turtles also occupy a variety of habitats during their various life stages, from nearshore waters to the remote open ocean [42,43]. To effectively manage sea turtle populations, accurate abundance estimates and characterization of their diverse habitat usage are crucial [44]. Understanding these ecological dynamics often requires gathering data repeatedly over many years, often decades, and across a wide range of geographic space [45]. Researchers customarily study habitat usage via satellite tracking [46] and conduct population assessments via routine nesting beach or foraging area surveys and tag/recapture studies [19,47]. These techniques are typically expensive and require special permits, especially when turtle capture is necessary for tagging and/or collecting biological samples. CS can, therefore, greatly benefit sea turtle research, as it helps attain spatiotemporal data that would otherwise be challenging for an individual research team to collect [48,49,50]. Indeed, CS is being used for sea turtle studies in many areas of the world [51,52,53,54].

### 1.3. The Southern California Sea Turtle Monitoring Project

In late 2012, a group of community members in Long Beach, in partnership with the local National Marine Fisheries Service (NMFS) office, the Aquarium of the Pacific (hereafter referred to as the “Aquarium”), the ecological consulting firm Tidal Influence, and the Los Cerritos Wetlands Authority, initiated a CS monitoring program, called the Southern California Sea Turtle Monitoring Project (also referred to as the “CS monitoring program”) to record simultaneous observations of green turtles once a month at select locations along the lower SGR where green turtle sightings had been prior recorded by NMFS biologists and local community members. At the core of this initiative was curiosity about the nature of green turtle occurrence and habitat usage in this very public, urban, and industrialized estuary, which at the time was only recently identified as a locality of consistent green turtle sightings through the collection of individual observations. An initial goal of this effort was to establish baseline data of green turtle activity near the power plants. At that time, both power plants were mandated to discontinue using OTC by 2020, and it was hypothesized that turtle activity could decline in the absence of the thermal effluent, similar to observations documented in San Diego Bay [55]. Today, this CS monitoring program is operated and managed through the Aquarium [56]. With both power plants now following an extended compliance schedule to discontinue their OTC systems [57], the Aquarium continues to operate the CS monitoring program to provide continuous data on green turtle habitat use and gain additional insights about their presence in the urban environment over time. 

Here, we report on the development of the Aquarium’s CS monitoring program, including observation data collected over nine full years of operation (2013–2021) and the environmental management implications gleaned from evaluating these CS data in the context of the surrounding wetland ecosystem and urban watershed. The primary goals of this study were to: (1) track the distribution of green turtle presence within the SGR monitoring site and determine the areas with the most green turtle activity over time, and (2) identify the underlying habitat features that potentially attract green turtles to these areas. Understanding green turtle hotspots along the SGR will aid in management efforts to reduce threats, and augment conservation and restoration efforts that can help sustain and expand the extent of the coastal estuary habitat in the region that is needed to promote the recovery of this threatened population. We intend this study to both inform sea turtle management and to highlight the utility of this CS program for monitoring the SGR green turtle population. This is the first multi-year dedicated study of the SGR green turtle population and the first application of a CS approach to monitoring green turtles in this region. 

## 2. Materials and Methods

### 2.1. Study Site

The SGR is an urban waterway that flows 93 km (58 mi) southward from its headwaters in the San Gabriel Mountains through 19 cities of the Los Angeles and Orange Counties of Southern California, USA [58,59,60]. The SGR’s mouth empties into the Pacific Ocean between the cities of Long Beach and Seal Beach (33°44′33″ N 118°06′56″ W, Figure 1). Approximately 6 km (3.7 mi) of estuarine habitat bordered with rocky levees extends from the mouth northward, at which point the SGR becomes concrete-lined for flood control and water conservation [58]. The benthic habitat of the tidal portion is non-vegetated, with a mix of soft and hard bottom covered in rubble, engineered structures (bridge pilings, cement discharge structures, and runoff outflows), debris, and intertidal sand beaches [61]. This habitat supports a diversity of marine benthic biota, including fish, polychaetes, annelids, mollusks, arthropods, and algae [62]. 

Green turtle monitoring sessions through the Aquarium’s CS program were conducted at 10 shoreside observation stations located along a 2.4-km (1.5 mi) stretch of the lower SGR (Stations 1 through 10, Figure 2). Each station is between approximately 50 and 60 m in length. Station 6.5 is an 11th station used specifically for training new volunteers prior to their official first session.

The Haynes Generating Station and Alamitos Energy Center (hereafter referred to collectively as “power plants”) are located on the SGR’s east and west banks, respectively, approximately 2.9 km (1.8 mi) from the mouth (Figure 2). The Aquarium’s 10 observation stations are strategically located near outfalls that discharge warm water effluent to the SGR from these two power plants, as well as both downriver and upriver of these locations. Stations 6, 6.5, and 7 are located directly above three outfalls from the Haynes Generating Station. Stations 7, 8, and 9 are located near the outfalls for the Alamitos Energy Center. 

The Los Cerritos Wetlands (LCW) straddle the SGR approximately 1.3 km (0.8 mi) upstream from where the river meets Alamitos Bay (Figure 2). This wetland complex comprises approximately 204 hectares (ha, 503 acres) of private and public property across both Long Beach and Seal Beach that were historically part of a much larger estuarine habitat [63]. Zedler Marsh, a 1.4-ha (3.5-acre) muted-tidal salt marsh within the LCW, directly borders the SGR on the eastern bank (Figure 2). This area has been undergoing habitat restoration since September 2009 [64], including trash removal and the planting of native wetland species. A 3-foot-diameter culvert connects Zedler Marsh to the SGR, which provides a direct path of tidal flow to sustain this ecosystem. In addition to the originally planted species, in recent years, a species of green alga commonly known as sea lettuce (*Ulva lactuca*) has been sighted growing in Zedler Marsh along with other intertidal algal species commonly found in Southern California coastal salt marshes (Figure 3). Downstream of Zedler Marsh, a smaller and more muted tidal marsh, Callaway Marsh, is also connected to the SGR via a 3-foot-diameter culvert. While the 0.3 ha (0.74-acre) Callaway Marsh does not possess the intertidal mudflats that promote algal growth, the marsh does host marine invertebrates for the SGR. Station 3 is located directly above the Zedler Marsh culvert, and Station 2 is located directly above the Callaway Marsh culvert (Figure 2).

### 2.2. Southern California Sea Turtle Monitoring Project Data Collection

Aquarium volunteers recorded green turtle observations on the first Saturday of each month (commencing circa October 2012). Although other sea turtle species that occur off the U.S. West Coast have been sighted offshore of California, to date, only green turtles have been recorded within the SGR by professional biologists, CS volunteers, and Aquarium staff overseeing the monitoring sessions. Monitoring sessions lasted for 30 min from 9:00 to 9:30 a.m. A minimum of 2–4 volunteers were assigned to each station, with one volunteer recording data on a paper observation log that included a station diagram (Figure 4). Throughout the monitoring session, volunteers recorded information each time a green turtle surfaced for air, which constitutes a “sighting” (also referred to as a “surfacing”). For each sighting, volunteers recorded location, surfacing time, and relative head size. Head size was reported as large (drawn as a circle, comparable in size to a softball or larger), medium (drawn as a square, comparable in size to a baseball), or small (drawn as a triangle, comparable in size to a golf ball or smaller) [65]. The presence of distinctive barnacles (e.g., *Chelonibia testudinaria*) or other markings visible on the head or body were also used as distinguishing features. Volunteers recorded other information in a notes box on the observation log, including other wildlife sighted (e.g., sea lions or jumping fish), weather patterns, and/or human activity. 

Volunteers assigned a numerical number to each sighting based on Aquarium protocols for identifying unique individuals. The purpose of this practice was to gain a general idea of the minimum number of green turtles that may have been present at a given station during a monitoring session. Each presumed individual sea turtle was numbered; a subsequent sighting was not numbered as a new individual unless volunteers could distinguish the sighting as a new individual via a simultaneous sighting (i.e., multiple turtles surfaced at once, meaning that the two surfacings must be separate individuals), a sighting with a different relative head size, or a sighting with a distinct physical feature (presence of barnacles, darker head color, etc.). 

Once the 30-min monitoring session concluded, the volunteer responsible for recording the data tallied the total number of sightings and estimated individuals, and delivered the observation log to the Aquarium’s volunteer coordinators overseeing the session. Data from the observation logs was later entered into a spreadsheet managed by Aquarium staff.

### 2.3. Observer Bias and Perception Bias

In order to minimize observer bias (e.g., data are recorded differently depending on the observer and their training and/or expectations of a certain outcome) [66,67], Aquarium staff followed standardized protocols to ensure that all volunteers were recording data in a consistent manner. New volunteers were required to attend one training session prior to their first official monitoring session. Training sessions took place at Station 6.5, which is located directly above an outfall that discharges warm water from the Haynes Generating Station (Figure 2) and has consistent green turtle presence every month. In addition to the initial training session, for their first three monitoring sessions, new volunteers were strategically assigned to observation stations with experienced volunteers so they had an opportunity to observe, ask questions, and confirm data recording protocols. At each monitoring session and once most of the volunteers arrived at the initial meeting location, Aquarium staff spent about 5–10 min reviewing data recording procedures to remind all volunteers of best practices and provided the opportunity to ask questions. Aquarium staff also minimized observer bias by rotating volunteers among observation stations to reduce their expectations as to whether they would see green turtles at a particular station or not. The only exception was for volunteers that are physically constrained from walking long distances; those volunteers were commonly assigned to Stations 4 through 6, which are closer to the initial volunteer gathering point. 

Perception bias (e.g., turtle is visible, but observers fail to see it because of varying conditions among observation stations) [68] is not considered to be influential in this study, as all observation stations are in close enough proximity that weather and water conditions (e.g., surface chop) are similar. The only exception is for the power plant stations where surface water is less smooth and more turbid compared to other stations because of discharge activity at the outfalls. Conditions such as higher surface chop would normally make turtle heads more challenging to sight compared to smoother surface waters; however, turtle heads tend to still be visible at the power plant stations because their dark color contrasts with the white foam created by the water turbidity.

### 2.4. Statistical Analysis

In order to determine the differences in the number of green turtle sightings among stations, years, and seasons, we developed a set of statistical models that describe the process of turtle sightings by the CS volunteers. The number of sightings at the i-th station on the k-th day was modeled by the multinomial distribution with the total number of observed sightings during the k-th survey day ($N_{k}$) and the proportion of the total for the station ($p_{k,i}$).
$$n_{k}∼multi\left(N_{k},p_{k}\right)$$

$n_{k}$ is a vector of length 10, which contains observed counts at 10 stations on the k-th day of observation. $N_{k}$ is the sum of all observed counts on the same day ($N_{k}=\overset{i=10}{\underset{i=1}{\sum }}n_{k,i}$). $p_{k}$ is a vector of length 10, which contains estimated proportions of sightings at 10 stations ($\overset{10}{\underset{i=1}{\sum }}p_{k,i}=1$). We compared models among five possible cases of the proportions (i.e., p vector): (1) sampling-day-specific, (2) season-specific, (3) year-specific, (4) year-month-specific, and (5) year-season-specific. The models were fit to the observed data using the Bayesian approach via JAGS [69] and R [70] with the jagsUI package [71]. The convergence of Marcov chain Monte Carlo was determined using the Rhat statistics [72]. The performance of the models was compared using deviance information criteria (DIC). The JAGS code for this analysis is available upon request.

## 3. Results

Here, we summarize the sighting/surfacing data collected by Aquarium volunteers during a total of 101 monitoring sessions conducted from 2013 to 2021. Results and analyses on the estimated number of individuals and head size will be reported in a separate manuscript. 

### 3.1. Sightings by Year

From 2013 to 2021, the four stations with the highest cumulative means (i.e., annual sightings averaged over all nine years of data collection) were: Station 3 (mean = 152 sightings/year; SD = 109.9), followed by Station 2 (mean = 78 sightings/year; SD = 45.2), Station 6 (mean = 71 sightings/year; SD = 34.6), and Station 7 (mean = 57 sightings/year; SD = 35.7). Cumulative means are provided in Table 1 and depicted in Figure 5. 

In all years, Stations 2, 3, 6, or 7 ranked first or second for highest total annual sightings count. Station 3 consistently had the highest number of total sightings in each individual year except 2014 and 2021; Station 6 had the highest number of annual sightings in 2014 and Station 2 had the highest number in 2021 (Table 1). We note that in 2013, 2014, and 2019, select observation sessions were canceled due to unsafe weather conditions, and in 2020, two observation sessions were canceled due to the COVID-19 pandemic, which reduced the total annual sightings in these years relative to other years. To account for this, we calculated and presented sightings per session by year as a proxy for catch per unit effort (CPUE) in Table 1. 

From 2016 to 2017, almost all stations experienced between a 40 and 94% increase in total annual sightings. From 2016 to 2018, the number of sightings at Station 3 exceeded the number of sightings at the stations with the next highest number of sightings by over 100 sightings (137 more sightings than Station 7 in 2016, 254 more sightings than Station 6 in 2017, and 104 more sightings than Station 2 in 2018). 

### 3.2. Sightings by Month

From 2013 to 2021, months in the summer and early fall seasons (June to October) typically yielded higher numbers of sightings, and months in the early winter/spring seasons (November to May) typically yielded lower numbers of sightings (Figure 6). This pattern was expected, as coastal temperatures off Southern California are higher during the summer months. 

### 3.3. Statistcal Analysis Results

Because of the modeling approach (i.e., multinomial model), sampling days that did not have data from one or more stations were eliminated. Among the five models that were fit to the remaining data, the fifth model (year-season-specific proportions) was considered best (lowest DIC value, Table 2). The variability among years was large, but in general, stations 2 and 3 had higher mean proportions of sightings among the 10 stations (Figure 7). In the recent years (2019, 2020, and 2021), however, Stations 2 and 3 did not show higher mean proportions than other stations. Station 6 also had a higher mean proportion of sightings relative to other stations in multiple years. 

## 4. Discussion

CS studies can significantly contribute to research in ecology and conservation, including the management of threatened species [50]. This study provides a multi-year baseline dataset on the presence of green turtles in the SGR that would otherwise not have been attainable were it not for the Aquarium’s CS monitoring program. This research constitutes one of the initial studies monitoring green turtles in the eastern Pacific Ocean using a CS approach (the other being Hanna et al. [54]).

### 4.1. Novel Discovery of SGR Green Turtle Habitat Association with the Los Cerritos Wetlands 

The most notable pattern identified in the CS data is the consistent high number of sightings at the stations with culvert connections to the LCW, a characteristic that all other stations lack. Station 3, which yielded the highest cumulative average of annual sightings (mean = 152 sightings/year; SD = 109.9, Table 1, Figure 5) and showed a higher mean proportion of sightings relative to other stations in most years (Figure 7), is located directly adjacent to Zedler Marsh of the LCW. A growing body of research shows a strong habitat association between green turtles and wetland ecosystems for forage opportunity [73]. However, the strong association with Station 3 and Zedler Marsh is still surprising, as there is a total absence of seagrass in the SGR [61] and the dimensions of the culvert and rebar debris prevent the turtles from directly accessing other forage foods in Zedler Marsh. Despite the lack of physical access to the LCW, there is regular tidal flow between the SGR and Zedler Marsh via the culvert, which would provide an opportunity for nutrient exchange. Given the clear presence of *U. lactuca* in Zedler Marsh (Figure 3), which is a common food source for green turtles [74,75,76], we hypothesize that *U. lactuca* debris, as well as other algae and small invertebrates, are dispersed to Station 3 via tidal exchange through the culvert, and, therefore, attract green turtles to this section of the SGR. Although green turtles typically feed on benthic food sources [77], they have been documented feeding in the midwater column as well [78], which could explain their higher activity at areas with potential forage debris. In addition to foraging opportunities, green turtles may be attracted to warmer waters discharged to the SGR from Zedler Marsh. Based on unpublished water temperature data collected by Tidal Influence, the water in Zedler Marsh’s shallow intertidal mudflat areas is slightly warmer than the water in the SGR. This is likely due to the dark sediment in Zedler Marsh warming the tidal water during high tide events [79]. Perhaps green turtles can sense these subtle temperature differences, choosing the areas with warm water due to energetic advantages offered [18,80,81].

Consequently, similar justifications may explain the consistent high number of sightings at Station 2, which is the station that yielded the second highest cumulative mean of annual sightings (mean = 78 sightings per year; SD = 45.2, Table 1, Figure 5) and also showed a higher proportion of sightings in most years (Figure 7). Station 2 has a direct connection to the LCW via the Callaway Marsh culvert. Although there is no observed algal growth in the area of Callaway Marsh adjacent to Station 2, green turtles are likely attracted to small invertebrates that flush out of the LCW and into the SGR via the Callaway culvert [79]. 

The high green turtle activity near the LCW stations corroborates other local observations. Since 2015, NMFS has solicited sighting reports of sea turtles from the public, which has resulted in numerous green turtle sighting reports in estuarine ecosystems along the Southern California coast, including Mugu Lagoon in Ventura, the Bolsa Chica Wetlands in Huntington Beach, and Agua Hedionda Lagoon in Carlsbad [20]. In addition, since 2008, green turtles have been consistently observed and studied in a restored estuarine basin with abundant eelgrass beds in the neighboring Seal Beach National Wildlife Refuge, located approximately 4.6 km (2.9 mi) east of the SGR’s mouth [30,82]. 

Due to both urban and agricultural development, California has lost approximately 90% of its coastal wetlands [83]. Globally, degradation of nearshore and estuarine habitats in general is accelerating [84,85,86,87], which will likely impact the suitability of these habitats for green turtle populations. Our observations of high green turtle activity near the LCW reinforce the ongoing necessity for habitat restoration in the LCW, as well as the other remaining tidal wetland areas along the California coast, to maximize the quantity, quality, and value of habitat available in Southern California for this recovering population of green turtles.

### 4.2. Sighting Trends over Time

There was an evident increase in annual total sightings across almost all stations from 2016 to 2017 (Table 1). Additionally, Station 3 yielded particularly striking results from 2016 to 2018 when the annual total of sightings exceeded all other stations by over 100 sightings (Table 1). To our knowledge, no localized activity in the LCW or SGR can explain this remarkable escalation in sightings beyond the possibility of an increased preference of this area by individual turtles. However, on a more regional scale, a marine heatwave in the northeast Pacific Ocean, commonly referred to as the “Blob,” brought anomalously warm sea surface temperatures to the coastal zone off California in 2014–2016, causing many marine species to shift northward of their typical geographic ranges [88,89,90,91]. One study reported that elevated temperatures were observed for longer in California coastal estuaries [92], which are comparable habitats to the SGR/LCW estuarine ecosystem. As such, it is possible that warmer waters from this marine heatwave attracted more green turtles than typical to the SGR during its tenure. In addition, these unusually warm ocean temperatures stunted seasonal upwelling activity, which consequently reduced nutrient availability and caused the collapse of important marine foraging ecosystems, including kelp forests [93,94]. Although not specifically documented along the California coast, the Blob likely damaged Southern California seagrass ecosystems, similar to what has been recorded for seagrass ecosystems affected by marine heatwaves in other parts of the world [95,96,97]. As damaged West Coast marine ecosystems recovered in the years following the Blob, green turtle attraction to Station 3 in the SGR for forage opportunity may have been magnified to compensate for reduced forage opportunities elsewhere along the coast. 

Notably, the annual total of sightings across all observation stations acutely decreased in 2020 and 2021 (Table 1). Additionally, the mean proportion in sightings at Stations 2 and 3 decreased from 2019 to 2021 (Figure 7). The decrease in annual sightings in 2020 can be partially explained by the fact that two monitoring sessions that year were canceled, which would have decreased the total annual sightings relative to other years. However, the annual total sightings in 2021 exceeded 2020 by only 21 sightings and all 12 monitoring sessions were conducted in 2021. The 2020–2021 sightings decline could be reflective of a reduction in warm water effluent discharge, as both power plants began decommissioning some of their generating units in 2020 [29,98]. The decline could also be related to the retreat of the Blob and the subsequent recovery of California coastal ecosystems, which would have eased any enhanced site-specific foraging attraction to Station 3 and any overall increase in sighting activity at other observation stations.

### 4.3. Confirmation of Thermal Refuge in the SGR

The stations with the third and fourth highest cumulative mean annual sightings were Stations 6 and 7 (mean = 71 and 57 sightings/year; SD = 34.6 and 35.7, respectively, Table 1, Figure 5), which are the stations located directly above the Haynes Generating Station outfall locations and near one of the Alamitos Energy Center outfall locations (Figure 2). Although not as consistently high as Stations 2 and 3, Station 6 also showed higher mean proportions of sightings relative to other stations in multiple years (Figure 7). Compared to other stations, green turtle sightings were much lower at Stations 8 and 9 (Table 1), which are located near the other two Alamitos Energy Center outfall locations. This is not entirely unexpected, as the Alamitos Energy Center has historically discharged a much smaller volume of warm water effluent than the Haynes Generating Station, with an average daily pump rate almost half that of Haynes (326 million gallons vs. 581 million gallons [29]). In addition, Stations 8 and 9 are not located directly above these outfalls, but in-between them, making the turtle presence less likely to be detected at each monitoring session. We expected to see the overall pattern of higher sighting numbers near the power plant outfalls, as Crear et al. [30] tracked the movement of 15 green turtles in the SGR via acoustic telemetry, concluding that turtles were most frequently detected near or downstream of the power plants’ warm water effluent during the winter, supporting the theory that the green turtles use the power plants’ warm water effluent as a thermal refuge. This corroborates the results of other local studies conducted near the South Bay Power Plant in San Diego that documented a similar association between green turtle presence and power plant activity [26,55,99]. In addition, the presence of sea turtles associated with warm water effluent from coastal power plants has been documented at other coastal locations in the eastern Pacific Ocean, including Chile and Brazil [100,101]. 

Although we were able to document changes in turtle activity near the power plants, it is challenging to identify the exact causes for these changes without additional and more granular environmental information for the SGR. A consistent time series of temperature data at each observation station may help determine whether a dynamic temperature profile throughout the study area corresponds to changes in green turtle activity. Water temperature data at Stations 6 and 7 would allow researchers to better correlate turtle observations to outfall activity at the power plants, including the Haynes Generating Station, which decommissioned half of its generating units in 2020 [98].

Forage opportunity in the thermal effluent near these stations may be an additional attractant. Although not confirmed with quantitative measurements, Torezani et al. [101] suggested that green turtle aggregations near a thermal effluent area of a steel plant in Brazil coincided with an abundance of green alga species. An updated bottom habitat survey of the SGR would be useful to determine if there is more algal growth at Stations 6 and 7. Additionally, knowing which stations lack forage foods (e.g., minimal algae) could explain the less frequent green turtle activity at certain stations (e.g., Station 10). Regardless of the primary attractant, we anticipate that green turtle presence may decrease in the areas near the power plants after their full discontinuation of OTC in 2023 and 2029 for the Alamitos Energy Center and Haynes Generating Station, respectively [57], similar to that observed after the decommissioning of the South Bay Power Plant in San Diego Bay [26,55,99]. 

### 4.4. Citizen Science as an Effective Tool for Monitoring the SGR Green Turtle Population

Data collection by CS volunteers represents an effective and notable first step toward developing a robust monitoring program for the SGR green turtle population. The SGR is an easily accessible study location in a highly populated urban environment, which provides a platform that is well-suited to recruit a large amount of volunteer participation and promote public awareness and involvement in sea turtle conservation [27]. Additionally, the monitoring area yields consistent turtle surfacing activity that can be observed and recorded from close-range, shore-side stations in a less invasive manner than traditional capture/tagging techniques. Furthermore, the Aquarium’s CS observation protocol has been identified as a preferred approach over other research methods trialed by Aquarium scientists. For example, the Aquarium has informally tested the use of drone and underwater video surveys. In the drone survey, pilots observed turtles both before drones were deployed and after they landed, yet the drones did not photograph a single turtle during flight [102], which is likely a result of the SGR’s high turbidity [103]. Turbidity in the SGR also affected the trial underwater surveys, which deployed remotely operated cameras that failed to document sea turtle presence due to poor visibility [102]. In Crear et al. [30], acoustic telemetry offered valuable movement data, but had significant limitations, including a small sample size of 15 turtles, and limited transmitter life during the course of the study. Both Crear studies [30,82] benefitted from CS monitoring program data to increase efficiency in capture times and locations, and to supplement data from traditional collection methods. In sum, the Aquarium’s CS protocol has proven to be the most successful research method for gathering presence/absence data over time on SGR green turtles. 

A growing body of research shows that diverse types of CS projects can produce accurate datasets comparable to professionals [104,105,106,107,108]. A data quality assessment of Aquarium CS data revealed low rates of error (mean error of 6%) when comparing CS-recorded data to corresponding NMFS/Aquarium-recorded data [109]. In that data quality analysis, Aquarium staff and NMFS biologists reviewed video recordings of CS monitoring sessions over the course of 14 consecutive monitoring sessions in November 2014 to December 2015 and compared results to the corresponding CS observation logs. Through this effort, Aquarium/NMFS staff identified and addressed the primary sources of CS error. Common factors leading to data error points were addressed through refresher trainings, updated observation tools, and mentorship pairings for new volunteers. 

Although traditional research techniques are necessary to gather information about the biological characteristics of the SGR population [30,82], the Aquarium’s CS monitoring program lends support to the notion that, in some cases, CS volunteers can more efficiently collect large datasets over time and space compared to an individual researcher or research team [48,49,50]. Populations of green turtles in San Diego Bay and the Seal Beach National Wildlife Refuge are regularly sampled by NOAA’s SWFSC through mark-recapture programs; however, very few green turtle hotspots in Southern California are suitable for these types of directed research programs due to logistical and funding constraints. Through the Aquarium’s CS program, volunteers have provided a cost-efficient and consistent time series of spatiotemporal presence/absence data that provide insight about how green turtles use this habitat area. The data collected from this CS program, in combination with other research initiatives, will serve as a baseline for long-term monitoring in terms of overall population presence and activity that can potentially serve as a local index of abundance, as well as a means to detect local changes in behavior and habitat usage patterns over time. Given the recovering trajectory of East Pacific green turtles and the increasing significance of Southern California coastal habitats to supporting their recovery, it is imperative to understand what constitutes favorable green turtle habitat, and identify what can be done to protect and augment those habitat features where possible. 

## 5. Conclusions

Our research results reveal key messages for green turtle management and the utility of CS as a research tool. First, the consistent and common presence of green turtles near the LCW contributes evidence to a strong habitat association between green turtles and tidal wetlands, which reinforces the need for protection and restoration of these highly degraded ecosystems along the U.S. West Coast. Second, the Aquarium’s network of CS volunteers has now provided more than nine years of monthly data, which has not only confirmed consistent year-round use of the SGR by green turtles and revealed the areas of the SGR with the highest green turtle activity, but has also proven CS as an effective research approach to monitor this resident population of green turtles. As more years of CS data are collected via both the Aquarium’s monthly monitoring sessions and a new photo identification component, we expect to be able to answer additional research questions regarding seasonal presence by station, as well as generate a minimum total estimate of individuals inhabiting the study area.

Although the East Pacific population of green turtles is still in a recovery phase, it is considered one of the most successful conservation stories of an endangered species in recent history. We expect to continue seeing increased sightings of green turtles along the U.S. West coast, and thus recommend the continued monitoring of the SGR green turtles and surrounding nearshore habitats to document trends in presence and distribution. We also recommend using the Aquarium’s CS protocol to expand public monitoring and involvement to other areas in Southern California with an uptick in public sightings reports (e.g., Bolsa Chica Wetlands in Huntington Beach and Agua Hedionda Lagoon in Carlsbad), especially when investigating long-term sea turtle activity and habitat usage. Last, we support continued efforts for wetland restoration along the California coast, as we expect that an increasing number of green turtles will recruit to these habitats in the future.

## Figures and Tables

**Figure 1 animals-13-00434-f001:**
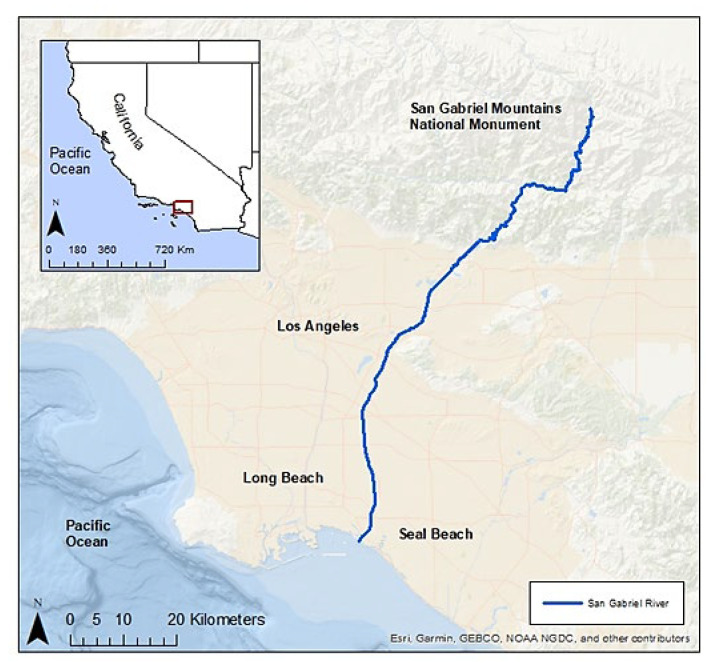
San Gabriel River, Southern California, USA.

**Figure 2 animals-13-00434-f002:**
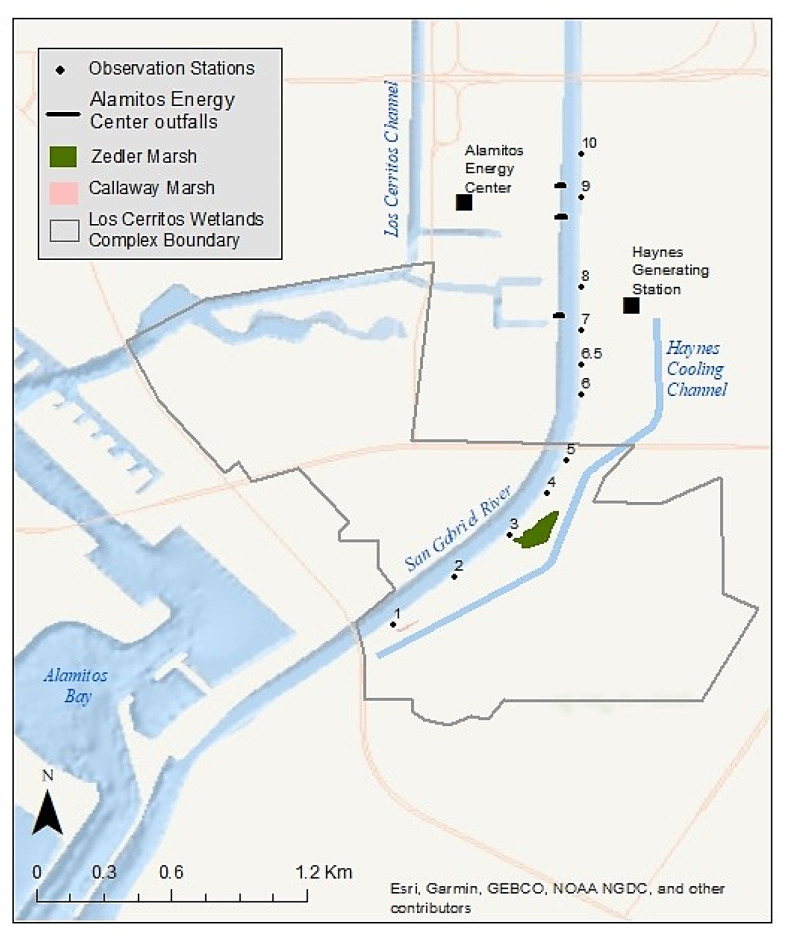
The Aquarium’s Citizen Science Monitoring Area. Stations 1 through 10 are observation stations. Station 6.5 is a training station; sightings data collected from this training location are not utilized in data analysis, to ensure data integrity. Stations 6, 6.5, and 7 are located directly above outfalls from the Haynes Generating Station. Alamitos Energy Center outfalls are labeled separately.

**Figure 3 animals-13-00434-f003:**
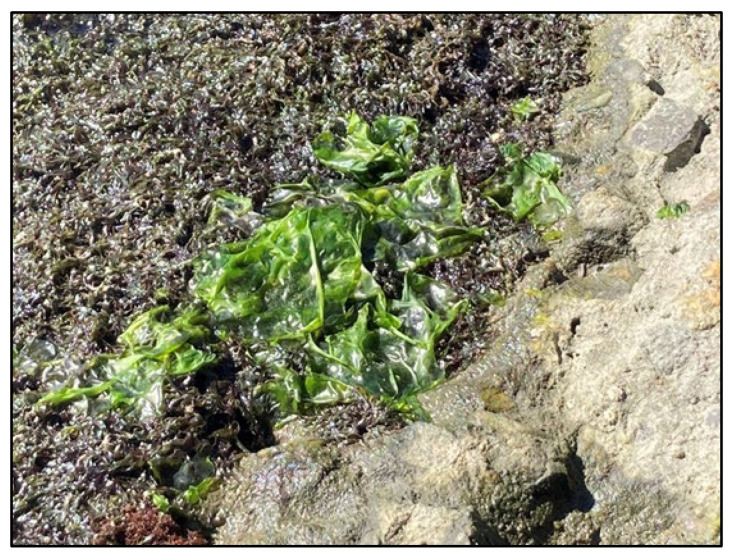
Image of *U. lactuca* at Zedler Marsh, documented on the wetland side of Station 3 by L. Massey and E. Zahn.

**Figure 4 animals-13-00434-f004:**
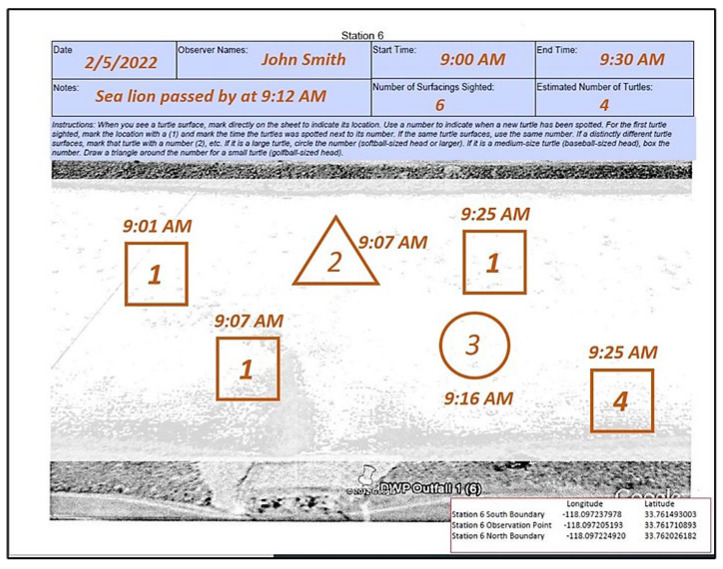
Example Observation Log.

**Figure 5 animals-13-00434-f005:**
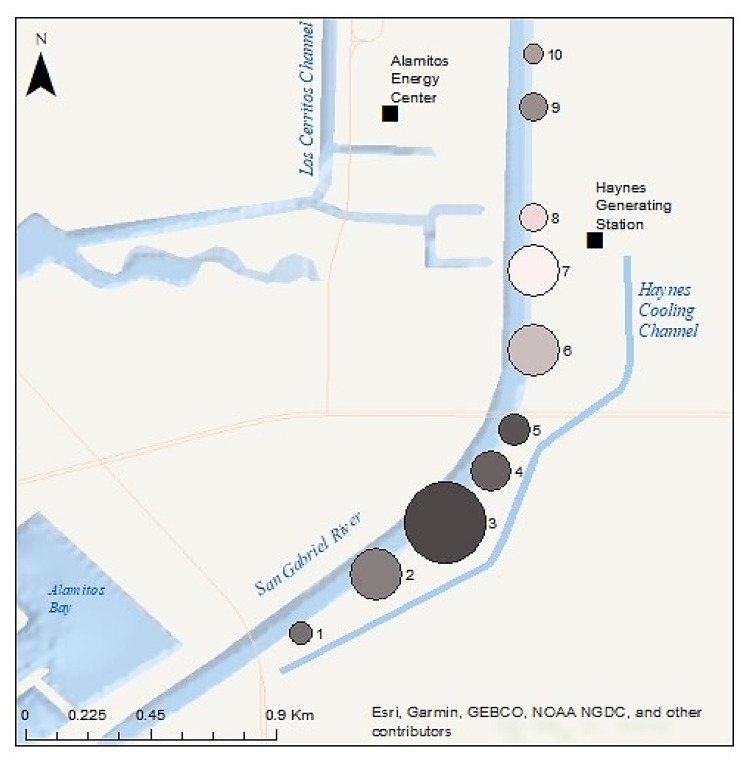
Cumulative Average Number of Turtle Sightings by Station 2013–2021. Larger circles indicate a higher cumulative average number of turtle sightings and vice versa. Averages displayed as circles are from Table 1.

**Figure 6 animals-13-00434-f006:**
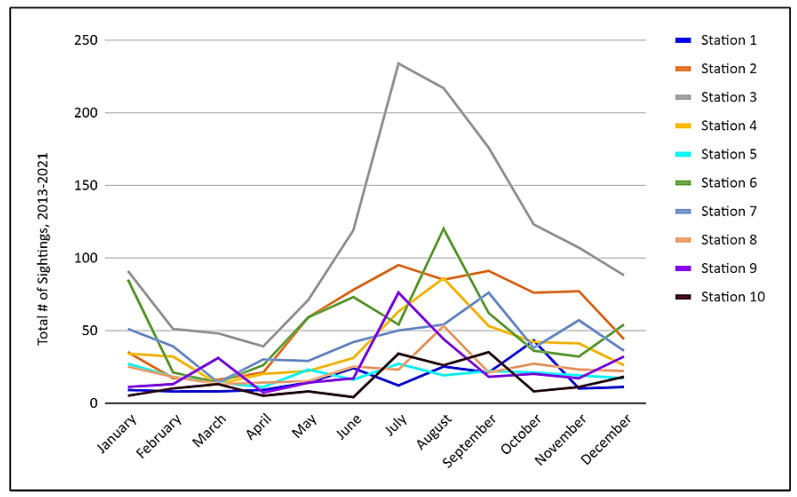
Sightings per Station by Month, 2013–2021.

**Figure 7 animals-13-00434-f007:**
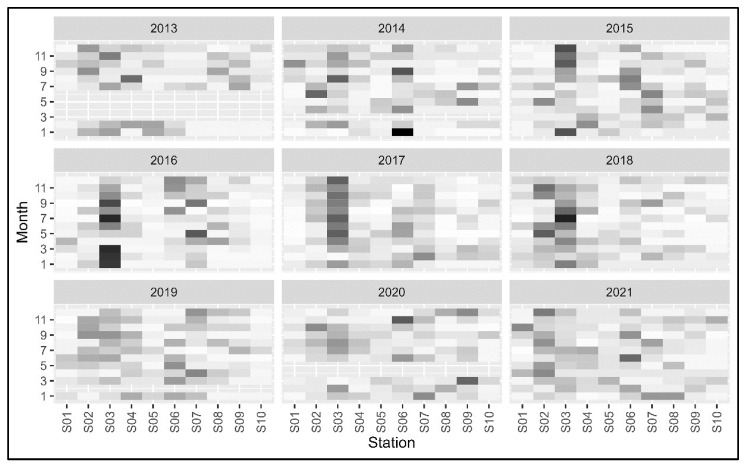
The mean proportions of sightings of green turtles at 10 observation stations along lower San Gabriel River, CA. Dark colors indicate higher values and vice versa.

**Table 1 animals-13-00434-t001:** Total Sightings per Station by Year, Annual Total Sightings by Year, and Cumulative Means per Station.

	2013	2014	2015	2016	2017	2018	2019	2020	2021	Mean Per Station for 2013–2021	Standard Deviation (SD)
**Station 1**	9	23	13	10	21	35	36	20	27	22	9.9
**Station 2**	38	54	28	28	126	125	142	62	96	78	45.2
**Station 3**	51	66	116	230	382	229	152	84	61	152	109.9
**Station 4**	38	26	32	23	76	72	116	39	42	52	30.5
**Station 5**	27	10	18	8	57	23	37	28	26	26	14.7
**Station 6**	12	99	68	63	128	48	103	51	67	71	34.6
**Station 7**	10	16	50	93	119	40	85	49	54	57	35.7
**Station 8**	13	15	20	33	34	32	58	28	46	31	14.5
**Station 9**	18	26	25	18	41	31	59	58	26	34	15.7
**Station 10**	7	15	19	3	49	24	27	20	15	20	13.3
**Annual Total**	223	350	389	509	1033	659	815	439	460	-	-
**CPUE**	25	32	32	42	86	55	74	44	38	-	-

**Table 2 animals-13-00434-t002:** Comparison of how models fit to the data when using the multinomial distribution. Rhat indicates the conversion of Markov chain Monte Carlo, where Rhat < 1.1 is considered to be acceptable. dDIC indicates the difference in deviance information criteria (DIC) value from the smallest DIC value, where the model with the smallest DIC is considered best.

Model	Rhat	dDIC
Multi-5	1.00	0.00
Multi-1	1.00	5.03
Multi-4	1.00	928.42
Multi-2	1.00	1673.27
Multi-3	1.00	2111.18

## Data Availability

Restrictions apply to the availability of these data. Data were obtained from the Aquarium of the Pacific and are available from the authors with the permission of the Aquarium of the Pacific.

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
