# Peer review of "Monitoring Green Sea Turtles in the San Gabriel River of Southern California"

_animals, 2023, doi:10.3390/ani13030434_

Round 1

Reviewer 1 Report

In this very interesting manuscript, Massey et al presented a beautiful set of data that suggests a potential reason for increase in population for green sea turtle around Southern California Pacific Coast. I think this study would have a lot of impact on efforts to protect endangered sea turtle population and also other aquatic animals. This is a very thoughtful manuscript and could be a very nice addition to MDPI-Animals. However, I have the following query for the authors and I would appreciate authors’ response on this, at least discussing them in the conclusion section of the manuscript.

1.    The effect that the authors observed, is it specific to the green sea turtle species? Would the other species of sea turtles favor the same conditions?

2.    The authors need to site some data about sea turtles in other regions of the world that may have conditions similar to San Gabrial river of Southern California.

3.    Is it possible to simulate similar conditions in laboratories to test how these environmental conditions are affecting the population growth? E.g. is it affecting sea turtle’s growth, breading or other developmental features?

4.    It would be important to know if these conditions are favorable for both male and female population of the sea turtles. Does the authors have any data for this?    

Author Response

We sincerely thank Reviewer 1 for their thoughtful feedback on this manuscript. We made the below changes through the lens of Reviewer 1’s comments and where necessary, explained if a request was not feasible for this study. We are happy to answer any additional questions Reviewer 1 may have.

  1.   The effect that the authors observed, is it specific to the green sea turtle species? Would the other species of sea turtles favor the same conditions?

No. Other species of marine turtles are not typically expected or known to reside in the nearshore/estuarine areas of Southern California. Leatherback, olive ridley, and juvenile loggerhead sea turtles are oceanic species in the eastern North Pacific ocean and have been observed in offshore California waters, but there have not been any recorded sightings of other sea turtle species in the San Gabriel River. We added text on page 6 in the Materials and Methods section to explain that only green sea turtles have been observed in the San Gabriel River. 

  1.   The authors need to site some data about sea turtles in other regions of the world that may have conditions similar to San Gabrial river of Southern California.

On page 12, in the paragraph titled “Confirmation of Thermal Refuge in the SGR,” we included a sentence that cites to green sea turtles congregating around power plant warm water effluent in other parts of the world, including Chile and Brazil: 

In addition, the presence of sea turtles associated with warm water effluent from coastal power plants has been documented at other coastal locations in the eastern Pacific Ocean, including Chile and Brazil [100-101]

We also cite to the green turtle population in San Diego Bay, a well-studied population which also exhibited congregation around warm water effluent discharge from the South Bay Power Plant prior to its closure. This example is discussed and cited in the Introduction section at the bottom of page 2 and in the Discussion section on page 12. 

  1.   Is it possible to simulate similar conditions in laboratories to test how these environmental conditions are affecting the population growth? E.g. is it affecting sea turtle’s growth, breading or other developmental features?

It is not feasible to simulate similar conditions in laboratory studies. East Pacific green sea turtles are a threatened species that are not typically subject to laboratory studies in the U.S. They are long-lived species that take decades to mature to reproductive age, which make developmental studies difficult to conduct. Also, there are complex associations between prey availability, estuarine water temperature, and environmental conditions across a large area of ocean between breeding/nesting areas of Mexico and foraging areas such as Southern California nearshore/estuaries, as well as a complex aggregation of threats to individuals across that large international scale, that cannot be simulated in a population growth context. Our manuscript is not evaluating the population dynamics of the species as whole; we are focused on the local dynamics of activity/presence. The information from the nesting beaches is what supports population growth of the species, and our data are at least congruent with a growing population in terms of increased activity/presence in a local area previously thought to be the fringe of the range.

  1.   It would be important to know if these conditions are favorable for both male and female population of the sea turtles. Does the authors have any data for this?    

No, sea turtle sex cannot be determined from visual observations of heads. It also cannot be determined from physical examination of juvenile turtles, of which many of these turtles that have been captured through direct research appear to be (NOAA’s Southwest Fisheries Science center unpublished, and Crear et al. 2016 and 2017). In order to determine sea turtle sex, blood must be drawn to perform hormonal analysis. This would require invasive capture and biological sampling techniques that are not within the scope of this research. On the foraging grounds, there is no literature that supports that males and females use nearshore estuarine areas differentially.

Reviewer 2 Report

Dear Author(s),

This study on the incidence of green turtles in the San Gabriel River reports that green turtles are more common around power plants, with seasonal peaks in summer and early fall seasons.

The results are based on observation only and the hypothesis of the research is not clear. For example, how can informing sea turtle management and at the same time emphasizing the usefulness of the CS program produce a scientific solution to a problem about sea turtles? You could convey this to the relevant institutions and organizations in the form of a report.

Also, what are the advantages and disadvantages of using only observational data? First of all, it is necessary to give information about when observation-based data should be used, what are the pros and cons, what questions will help to find it, and the main hypothesis of the study should be justified. 

My suggestion is that you define the question of the study well and emphasize how it will contribute to the life of the green turtle when you solve this problem.

The MM part of the study is not clear and unambiguous. Overdispersed should be taken into account when calculating observational data. Also, statistical approaches should be presented for Observer Bias and Perception Bias. What method was used for species identification? Are there other species in the area? Was the species identification of all observed turtles possible?

Is there a statistical difference between years and stations in terms of CPUE? Is there a statistical difference between years and seasons? Are there any trends as a result of the monitoring study over the years? 

 Also, it would be better to include water temperatures in the main data, as well.

Best Regards

Author Response

We sincerely thank Reviewer 2 for their thoughtful feedback on this manuscript. We made the below changes through the lens of Reviewer 2’s comments and where necessary, explained if a request was not feasible to address for this study. We are happy to answer any additional questions Reviewer 2 may have. 

This study on the incidence of green turtles in the San Gabriel River reports that green turtles are more common around power plants, with seasonal peaks in summer and early fall seasons.

The results are based on observation only and the hypothesis of the research is not clear. For example, how can informing sea turtle management and at the same time emphasizing the usefulness of the CS program produce a scientific solution to a problem about sea turtles? You could convey this to the relevant institutions and organizations in the form of a report.

We added text to the last paragraph of the Introduction section to better clarify the goals of the study and how the Aquarium’s CS monitoring program accomplishes these goals. We also added additional text on page 3 of the Introduction to delineate the initial goals of the Aquarium's CS program when it was first started from the intent of this study and its analyses. We emphasize the usefulness of the CS approach in order to highlight the suitability of CS for these particular research goals at this particular location and how it significantly benefits monitoring this threatened population of green turtles. Utilizing CS allowed for observations to occur simultaneously across a 2.4-km stretch of the lower San Gabriel River. Observations for such a duration and distance demonstrate the year-round presence of sea turtle activity in this novel habitat. This amount of spatial and temporal data collected through the Aquarium’s program would otherwise not have been feasible for an individual research team to collect. There are many logistical barriers to traditional research techniques, including the necessity for federal permitting to conduct any form of capture/tagging/biological sampling. 

We discuss the advantages of using CS as a research approach on page 2 in the Introduction section, specifically in the paragraph titled Citizen Science as a Tool for Sea Turtle Research. We discuss the more traditional research methods that were trialed and rejected for use at this study site on pages 14-15 in the Discussion section.

My suggestion is that you define the question of the study well and emphasize how it will contribute to the life of the green turtle when you solve this problem.

We added text to the last paragraph of the introduction section to better clarify the goals of the study. We discuss how the CS has contributed to the study in the Discussion on pages 14-15. 

The MM part of the study is not clear and unambiguous. Overdispersed should be taken into account when calculating observational data. 

We added a section on statistical analysis to the Materials and Methods section on page 8 and the Results section on pages 10-11. We developed a set of statistical models to determine the differences in the number of green turtle sightings among stations, years, and seasons. The resulting Figure 7 on page 11 shows the mean proportions of sightings over time. 

Also, statistical approaches should be presented for Observer Bias and Perception Bias.

This study was not designed to directly measure observer and perception bias. However, to address this comment, we added a relevant paragraph to the Discussion section on page 14 that describes a past study conducted to quantify the error rate between the observation data of citizen science volunteers vs. Aquarium staff/NMFS biologists. 

What method was used for species identification? Are there other species in the area? Was the species identification of all observed turtles possible?

We added text to the Materials and Methods section on page 6 to clarify that green sea turtles are the only marine turtle species that have been sighted in the San Gabriel River by professional biologists and Aquarium staff to date.

Is there a statistical difference between years and stations in terms of CPUE? Is there a statistical difference between years and seasons? Are there any trends as a result of the monitoring study over the years? 

A CPUE proxy was provided only to show how many total sightings occurred per session, taking into account the fact some years had canceled monitoring sessions. To address statistical difference over time, we developed a set of statistical models to determine the differences in the number of green turtle sightings among stations, years, and seasons. The resulting Figure 7 on page 11 shows the mean proportions of sightings over time. 

Also, it would be better to include water temperatures in the main data, as well.

We agree that water temperature data would be incredibly valuable, however we do not have river temperature data available for analytical use in this study. On page 12, in the section titled “Confirmation of Thermal Refuge in the SGR,” the second paragraph identifies the lack of river temperature data as a drawback in this study: 

Although we were able to document changes in turtle activity near the power plants, it is challenging to identify the exact causes for these changes without additional and more granular environmental information for the SGR. A consistent time series of temperature data at each monitoring station may help determine whether a dynamic temperature profile throughout the study area corresponds to changes in turtle activity. Water temperature data at Stations 6 and 7 would allow researchers to better correlate turtle observations to outfall activity at the power plants, including the Haynes Generating Station which decommissioned half of its generating units in 2020 [98].

Round 2

Reviewer 2 Report

Dear Authors,

This version of MS has been great compared to the previous version, both in terms of clarifying the purpose of the study and adding a statistical model. I enjoyed reading it.

Thank you very much for considering my suggestions.

Best Wishes